# A Knowledge-Grounded Task-Oriented Dialogue System with Hierarchical Structure for Enhancing Knowledge Selection

**DOI:** 10.3390/s23020685

**Published:** 2023-01-06

**Authors:** Hayoung Lee, Okran Jeong

**Affiliations:** School of Computing, Gachon University, 1342 Sujeong-gu, Seongnam-si 13120, Gyeonggi-do, Republic of Korea

**Keywords:** conversational AI, knowledge-grounded task-oriented dialogue system, knowledge selection, classification, named entity recognition, snippet ranking, negative sampling

## Abstract

For a task-oriented dialogue system to provide appropriate answers to and services for users’ questions, it is necessary for it to be able to utilize knowledge related to the topic of the conversation. Therefore, the system should be able to select the most appropriate knowledge snippet from the knowledge base, where external unstructured knowledge is used to respond to user requests that cannot be solved by the internal knowledge addressed by the database or application programming interface. Therefore, this paper constructs a three-step knowledge-grounded task-oriented dialogue system with knowledge-seeking-turn detection, knowledge selection, and knowledge-grounded generation. In particular, we propose a hierarchical structure of domain-classification, entity-extraction, and snippet-ranking tasks by subdividing the knowledge selection step. Each task is performed through the pre-trained language model with advanced techniques to finally determine the knowledge snippet to be used to generate a response. Furthermore, the domain and entity information obtained because of the previous task is used as knowledge to reduce the search range of candidates, thereby improving the performance and efficiency of knowledge selection and proving it through experiments.

## 1. Introduction

Research is actively underway on conversational Artificial Intelligence (AI), which aims not only to successfully mimic human conversations, but also to appropriately provide knowledge-based answers and actions to users’ questions [1,2,3,4]. A representative conversational AI is the Task-Oriented Dialogue (ToD) system, which focuses on providing the information needed by a given database or API and performing specific actions closely related to real life, such as flight and hotel reservations. It is widely used in our lives as a digital personal assistant or customer service bot, and such a typical ToD system is generally configured so that users ask questions and the system responds in a manner similar to frequently asked questions (FAQs) [5,6].

ToD systems are distinguished from social bots, which aim to provide satisfaction by enabling natural and smooth conversations with humans on various topics based on an open domain [1]. Therefore, it is important to generate responses using information about the topic of the conversation to better understand the meaning contained in a user’s speech to provide the right service or to respond fluently and accurately to a user’s questions. However, as most of the tasks requested through conversation work only in environments limited to a given range of databases or APIs, there is a disadvantage in that the interaction of the conversation is inefficient because no response is provided to requests outside the scope [7]. Accordingly, studies on knowledge-grounded task-oriented dialogue systems that utilize a wide range of knowledge bases derived from various data sources, including web pages, such as Wikipedia [8], or knowledge graphs [9] for response generation have recently been conducted. A knowledge-grounded task-oriented dialogue system consists of three main steps: knowledge-seeking turn detection, which is a step to determine if knowledge is needed in the user’s utterance, knowledge selection, which is a step to decide what knowledge to use, and knowledge-grounded generation to generate a response based on the selected knowledge [7].

In order to provide satisfaction to the user, the ultimate goal of the conversation system, the most accurate knowledge snippet must be selected from a large-scale external, unstructured knowledge base [6]. In addition, for the conversational turn classified as requiring knowledge in the multi-turn conversation between the user and the system, the knowledge-selection step is very important because the system must be able to select the most appropriate knowledge snippet to generate the right response. Therefore, to efficiently utilize related domain knowledge, such as FAQs and customer reviews, called unstructured knowledge, we intend to improve the performance of a knowledge-grounded ToD system by applying detailed tasks to the knowledge-selection stage.

First, to understand the context of the conversation with the user, it is necessary to proceed with a task that classifies the domain of the overall conversation based on the conversational history. Next, a task of extracting an entity that corresponds to each domain and can be a major keyword in response generation should be performed [10]. In addition, to obtain documents that can be used as background knowledge for response generation, it is possible to divide the appropriate snippet candidates into filtering tasks through the ranking step of knowledge snippets. Therefore, the knowledge-selection step can be subdivided into a hierarchical structure consisting of three levels of tasks: domain classification, entity extraction, and snippet Ranking. At this time, the predictive results of the first two steps are sequentially used for the last snippet-ranking task, which is used to reduce the search scope of knowledge candidates. If the dialogue system is configured based on the corresponding task, it can be expressed as shown in Figure 1.

In this paper, each task is successfully implemented by performing fine-tuning pre-trained language models according to the purpose of the task. First, in the domain-classification step, the model is designed to properly classify the problem among several domains present in the dataset through multi-class classification. The entity-extraction task is then solved by approaching token classification among the Named Entity Recognition (NER) problems to extract it with the right entity from the numerous entity lists that exist in the determined domain. We train the model to successfully conduct token classification by applying IOB (Inside–Outside–Beginning) tagging to input embeddings, utilizing the information of each entity of the external unstructured knowledge in the configured conversational turns. Additionally, by conducting an experiment by setting conversational turns differently for the above two steps, we will also find out the number of turns in conversational history that have a major influence when generating a response. The last snippet-ranking task uses the pre-trained language model as the ranking model to select the appropriate knowledge snippet candidates to generate a response to the user query. In addition, to increase the training difficulty and selection performance, negative sampling techniques are used to generate top-k candidate lists for each conversational turn that requires knowledge [11].

In this paper, our major contributions are as follows:The knowledge selection step of knowledge-grounded task-oriented dialogue modeling is subdivided into hierarchical structures of domain classification, entity extraction, and snippet ranking, which could be successfully performed by fine-tuning each task using pre-trained language models;The domain-classification task approaches multi-class classification and configures various conversational turns to successfully determine the most appropriate domain among multiple domains;The entity-extraction task approaches the NER problem with IOB tagging to extract the entity contained in the conversational turn and performs token classification by using the domain-classification results as knowledge;The snippet-ranking task is trained to construct the snippet candidate list to be used for response generation by using the pre-trained language model as a ranker model, which improves performance by applying the negative sampling technique.

The remainder of this paper is organized as follows. In Section 2, we summarize related work. In Section 3, we describe the dataset for the knowledge-grounded task-oriented dialogue system and our proposed method for enhancing knowledge selection. Section 4 shows the results of our validation experiments, and Section 5 concludes the paper.

## 2. Related Work

Conversational AI has been developed to provide free chit-chat conversations or appropriate information using knowledge to answer users’ questions. As knowledge is in a large-scale unstructured structure [12], to implement an intelligent open-domain dialogue agent, [8] acquired knowledge from Wikipedia web pages to form a dataset, and [13] proposed a dataset that conducts conversations on eight broad topics, training a knowledge-grounded social bot with an encoder–decoder conversational model. In addition, [14] proposed Multi-Domain Wizard-of-Oz (MultiWOZ), which covers 10k dialogs and topics to train ToD systems, and [9,15] proposed a common-sense knowledge-based conversational model that was configured and utilized as a knowledge base in the form of a knowledge graph.

In order to efficiently apply the configured knowledge base to a task-oriented dialogue system, several studies are being conducted based on pre-trained language models. In [16], a single language model trained from all sub-tasks through a simple TOD system was integrated and used as a single sequence-prediction problem, which improved the performance of dialogue state-tracking through Generative Pre-trained Transformer 2 (GPT2). In addition, [17] proposed TOD-BERT, which overperforms Bidirectional Encoder Representations from Transformers (BERT) in intent recognition, dialogue state tracking, dialogue act prediction, and response selection by integrating user and system tokens into language modeling and training with ToD datasets. On the other hand, unlike a model that generates a conversational interaction based on a general scenario, [18] utilized dialogue context and knowledge documents as an encoder–decoder model. Additionally, [19] is being efficiently used to perform response generation based on unlabeled dialogues by utilizing the optimizing knowledge selection step based on unsupervised learning in related knowledge documents.

Additionally, to successfully implement a knowledge-grounded system, various Information Retrieval (IR) techniques are used to bring in the desired knowledge. In [20], keywords were extracted from a query dataset through Term Frequency–Inverse Document Frequency (TF-IDF) to be searched for the most relevant reply, and [21] is being used to generate relevant information for user responses. In addition, in [22], to use external knowledge to select responses in a knowledge-aware retrieval-based chatbot system, they presented a document-grounded matching network that achieved state-of-the-art. Additionally, [23] proposed an end-to-end process that directly learns ranking scores using neural networks, and [19] successfully implemented knowledge generation and knowledge generation based on a pre-trained language model.

There are several successful IR techniques for utilizing external knowledge, but NER exists to extract the desired entity from the text and use it with the pre-trained language model. Here, the named entity is a word or phrase that clearly identifies an item in a set of items with similar properties and generally exists as an organization, person, location name, etc. That is, NER is a process of finding named entities in text and classifying them as categories of predefined entities. Therefore, NER acts as an important preprocessing step for various applications, such as information retrieval, question answering, and machine translation [24]. A method of learning NER is first clustering, which is a technique of extracting named entities from a classified group based on text similarity with a general unsupervised learning approach. In other words, instead of utilizing a labeled dataset, it is solved by applying the idea that the reference of the named entity can be inferred using statistics calculated from a large corpus and syntactic knowledge [25]. On the other hand, in the case of feature-based supervised learning approaches given an annotated data sample, it is applied to multi-class classification or sequence-labeling tasks and is also trained to recognize similar patterns from data using multiple machine-learning algorithms [26,27,28].

However, when NER is applied to deep learning, non-linear mapping from input to output can be generated to learn much more complex and intensive features from the data than a linear model. In addition, it is effective in learning useful representations and basic elements from raw data through deep-learning-based models, so excellent performance can be expected [29,30,31]. Therefore, in this paper, we successfully perform an entity-extraction task, the second task of knowledge selection, by utilizing the deep-learning-based NER with the neural language model. Therefore, Part-Of-Speech (POS) tagging was integrated and used to consider word- and character-level embedding as distributed presentations for input. In other words, by predicting the tag for the token of the input sequence as one of the named entity types B-(begin), I-(inside), and O-(outside), we trained our proposed model to detect the boundary of the entity and then classified the range of the detected text as the type of entity.

Therefore, based on the studies that have been conducted so far, this paper intends to extract desired information, such as the domain and entity, using pre-trained language models for the knowledge-selection stage and construct a ranking model by applying the negative sampling method.

## 3. Proposed Method

Knowledge-based dialogue systems not only need to understand what users are saying well, but also generate appropriate responses based on available internal and external knowledge. Accordingly, the main challenge of knowledge-grounded task-oriented dialogue systems is the selection of the most appropriate knowledge snippet among large-scale knowledge document candidates, considering the knowledge-seeking turn conversation. In other words, the direction of the response generated by the conversation system itself is completely different depending on what knowledge the conversation system selects. Therefore, this paper focuses on the knowledge-selection task, which has a significant impact on performance among the three stages in the task-oriented dialogue system that utilizes external unstructured knowledge as a knowledge base presented in [7].

In this paper, the knowledge-selection stage is subdivided into three hierarchical structures: domain classification, entity extraction, and snippet ranking, and the overall baseline architecture is shown in Figure 2. First, to understand the user’s speech more systematically and make decisions, the domain is determined through the conversational history, and then the entity for each domain is determined. Subsequently, among the several knowledge snippets listed by entity, it is ranked to determine the appropriate snippet to be used for generating responses. Accordingly, a detailed three-step implementation method will be described.

### 3.1. Domain Classification

To select the appropriate knowledge candidate for conversation generation, it is necessary to first understand the context of the dialogue from the conversation history to be used as the input for this task. Therefore, it is important to identify the appropriate domain, which is the context of the ongoing dialogue about the conversation turn that requires knowledge. Although it may be considered relatively minor compared with the other two tasks performed according to the step, it is essential because the entity and knowledge snippet themselves that can be selected in the next task may vary depending on the domain determined as the result of this task. Furthermore, the computational complexity and memory usage of the following tasks can be lowered by reducing the search scope of knowledge candidates based on the domain classification result, the output obtained by the execution of this task.

In this paper, there are five domains consisting of a dataset used to achieve knowledge-grounded task-oriented dialogue modeling: hotel, restaurant, taxi, bus, and attraction. Therefore, this domain-classification task is solved by multi-class classification [10], which is classified as one of the above five domains, and fine tuning is performed by finally adding a linear layer for the multi-class classification task, which has shown excellent performance in natural-language-processing problems. At this time, as an input for model training, we want to classify the appropriate domain using only the convergent history of the dataset, which is shown in Figure 3. In addition, experiments are conducted together to identify where important information is mainly located to understand the context of the conversation. Assuming that the key information needed to generate a response is mainly located in the last sentence and near it in a multi-turn conversation between the user and the system with up to 15 turns, the experiment was conducted by configuring the number of conversational turns differently from the bottom to {1, 3, 5, 10, 15}.

### 3.2. Entity Extraction

Following the domain-classification task is a step to determine exactly what entity is being talked about in the conversation history. In general, looking at conversations between people, users tend to ask questions by including the information they want in their speech, so the entity we want to obtain in the task is included in the conversation with a high probability. Therefore, we design the problem with NER, which clearly identifies an item named an entity from multiple sets of items with similar properties. In other words, to extract where the entity is located from the history of the conversation with the user, token classification [32,33] can be performed using POS tagging together to predict the tag for the token of the input sequence as one of the types of named entity, B-(begin), I-(inside), and O-(outside), which is represented in Figure 4. After training to detect the boundary of the entity, the range of detected text is classified as the type of entity. In addition, the domain information obtained as a result of the domain classification conducted above is used as knowledge for extracting entities and is shown in Figure 5.

Therefore, we would like to configure the input of the model by adding domain information along with the conversational history. Accordingly, input embedding is organized into <CLS> Entity <SEP> history by adding the <SEP> token, which is a special token, to distinguish information between the two categories. Here, like the domain classification task, the number of turns in the conversational history is set differently to check the results. However, as there is no entity in the taxi and train domain configured at this time, an entity-extraction task is only performed on hotel, restaurant, and attraction among the five domains.

### 3.3. Snippet Ranking

To efficiently use external knowledge in an intelligent dialogue system, it is important to select an appropriate snippet and use it as knowledge. Even if the appropriate domain and entity have been identified for conversational turns requiring knowledge through previous tasks, the number of knowledge snippets that exist in this regard can range from tens to hundreds or more. Therefore, in order to decide which of the many snippets to use as knowledge to generate responses, it is necessary to finally classify the appropriate snippets as candidates by calculating and ranking the relationship between the conversational history and each knowledge snippet [34]. In this paper, the knowledge in which the domain and entity match the corresponding conversation in the entire external unstructured knowledge base is set as a positive sample. In addition, the relevance function is trained to classify positive samples from negative samples in the entire knowledge base, and the pre-trained language model is used as a ranking model.

However, currently, there are more than 2900 knowledge snippets in the training and validation sets [14], so encoding and using all of them for training is inefficient in terms of computational complexity and memory usage. Therefore, negative samples are constructed by reducing the range of candidates to be selected as snippets by using the domain and entity obtained from the previous task as information. Using this as knowledge, we intend to improve the performance of the ranking model by configuring input embedding with the conversational history and conducting training, as shown in Figure 6. Therefore, for performance comparison, training is conducted using negative samples in various ways [35] through the methods configured below.

All: Train using all documents as candidates;Positive: Only snippets corresponding to the domain and entity matching the conversational turn are used as candidates;Random: Select negative samples randomly as the same number as the positive samples to configure the candidate with the positive samples;In-domain: For this conversational turn, a negative sample is constructed as many positive samples as the number of positive samples by randomly selecting the snippet so that the domain matches but the entity is different, and used as a candidate with the positive sample. At this time, in the case of taxi and train, where the entity does not exist in the domain, it is configured in the same way as ‘random’.

Except for the all and positive methods, negative samples composed of the above method are used together with positive samples based on the knowledge provided for each conversation. In addition, performance is improved by uniformly sampling negative samples at a ratio of 1:1 to positive samples only when training the model. When evaluating the model, all knowledge snippets are used as candidates to make sure that the appropriate knowledge snippets are selected well.

### 3.4. Dataset

In this paper, to successfully train the task-oriented knowledge-grounded dialogue system, we conducted an experiment on the knowledge-selection stage that has the most significant effect on response generation. To this end, the MutliWOZ 2.1 dataset used in DSTC9 Track-1 is used [14,36], which is a version of MultiWOZ 2.1 augmented using conversation data between tourists and clerks based on San Francisco tourism information. Therefore, in addition to the existing API, questions outside the API coverage are inserted with external knowledge access. To evaluate the generalization capability of the ToD system, the test set includes conversations related to the new domain. Detailed statistics for these configured datasets are summarized in Table 1.

The dataset contains information on the knowledge-seeking turn as a binary label for each dialogue, and knowledge snippet and ground-truth response information exist for true. The training, validation, and test sets consist of conversations with 71,348, 9663, and 4181 multi-turns, respectively, of which the ratios of knowledge-seeking turns were 26.9% as 19,184, 27.7% as 2673, and 47.4% as 1981, respectively. There are five domains: hotel, restaurant, taxi, train, and attraction, where the attraction domain is included only in the test set. In addition, as the entity only contains information regarding the hotel, restaurant, and attraction domains among the above domains, it is necessary to use the domain information obtained as the result of the domain-classification task to exclude taxi and train from the corresponding task. As the knowledge-selection part is performed on the dataset, the experiment is conducted only on data whose target is ‘True’, which is the binary label existing in each dialogue.

## 4. Experiments

The domain-classification task is a multi-class classification task that determines which domain belongs to among multiple classes, i.e., domains. As the entity-extraction task that follows is also solved by token classification, it is a task that classifies the conversation as a token and then finds the location of the appropriate entity. Therefore, experiments were conducted using pre-trained language models that perform sufficiently well in natural language processing tasks, such as BERT [37] and a distilled version of BERT (DistilBERT) [38]. As representative experimental parameters, the learning rate was set to 1e-5, epsilon was set to 1e-8, and the experiment was conducted through the Adam optimizer.

Recent studies claimed that most of the information needed to generate the next response is in the user’s last utterance and constructed input embedding of the model using only the last utterance of conversational history. However, as the data of each conversation consisted of up to 15-turn dialogues, we examined the classification results that differed for each case by configuring and experimenting with a conversational history to use as the input for the above two tasks. At this time, we wanted to use a lot of information by setting the max token length for history to 512, which was relatively large, and model training was conducted by setting the epoch to 10. As both tasks were ultimately classification tasks, the performance was evaluated using the precision, recall, and f1-score as evaluation metrics.

On the other hand, the snippet-ranking task is to rank snippets to achieve the top-k candidates to select the knowledge that can be used to generate responses, because it is most relevant to the context of an ongoing conversation among several knowledge snippets. In this paper, we implemented the ranker model using XLNet [39], which is known to outperform the GPT2 model [40] implemented in the baseline model. Regarding the configured experimental parameters, the learning rate was set to 6.25e-05, Adam epsilon was set to 1e-08, and the max token length was limited to 128 for knowledge and conversational history, respectively. Through the Adam optimizer, the network conducts training in the direction of minimizing cross entropy loss between the model’s output and ground-truth label. To verify the performance of the proposed model, a comparative experiment is conducted by configuring and training the same as the basic experimental environment of the baseline models to be compared. 

To check the performance of the model, MRR@5, recall@1, and recall@5 were used as evaluation metrics.

### 4.1. Experiment for Domain Classification

To solve a problem using a deep-learning model, it is necessary to provide an appropriate input representation. At this time, input representation is learned by word embedding, and BERT can extract contextualized vector representation [41], which is often used to solve multi-class classification problems. In addition, DistilBERT is a language model that reduces size, but improves speed by applying knowledge distillation techniques to BERT while preserving performance [38]. Unlike the use of knowledge distillation in task-specific models in general, DistilBERT was used in the pre-training process, showing good performance in various tasks, such as BERT. Therefore, the results of multi-class classification through these BERT-based models are shown in Table 2 below.

Overall, when comparing the results between models, in the case of DistilBERT, the higher the number of dialogue turns, the better the results were for all metrics. Looking at the experimental results of setting different # dialogue turn, the overall configuration of 5 turns from the last utterance was not much different from the case of 15 turns, which was the result of using all the conversational history. According to this, most of the information that has an important influence on response generation exists within five turns from the last utterance. In addition, the results for one turn, the result of training using only the last utterance, did not show a relatively large difference compared with the other results. Although the overall metric value was lowered by a narrow margin, it could also be seen that the last utterance played the most important role in the next response.

Additionally, looking at the overall experimental results, it was usually around 70%, and among the domains of the MultiWOZ dataset where the experiment was conducted, the attraction did not exist in the training and validation sets, only in the test set. Therefore, in the case of DistilBERT, which was the highest-performing model, the evaluation was conducted with a test set. The classification results for each class are examined in Table 3, and there is no classification at all for attraction. However, as the rest of the domain is being classified successfully, the overall metric value is slightly lower due to the attraction.

### 4.2. Experiment for Entity Extraction

Named entity recognition is classified as one of the types of tags configured using POS tagging, which corresponds to multi-class classification. Therefore, it was implemented using the BERT-based pre-trained language model in the same way as the domain-classification task. To extract the entity included in the conversational history, it is necessary to first check whether the conversational history configured according to the set number of dialogue turns contains the entity, and to configure embedding by applying IOB tagging. Therefore, the percentage of the configured domain that contains the identity depending on the number of dialogue turns for the training, validation, and test sets can be expressed, as in Table 4. At this time, looking at the case of the training set, which constituted the entire conversational history, the conversation included entities at a rate of about 95%.

In the previous information on domain, even if the conversational history was composed of only one turn, there was no significant difference from the entire history, whereas, in terms of entity, the inclusion ratio was very low. As a result, looking at the results of entity extraction in Table 5, in the case of one turn, if the entity was not included, the classification accuracy was much higher than that of other turns because it was separated by O for all tokens. On the other hand, when the conversational history was composed of 5 turns, there was no significant difference in the entity inclusion ratio compared with the case where the conversational history was composed of 10 turns; rather, it can be observed that the token-classification performance was low in the extraction result. Therefore, the longer the conversational history, the more noise can act as the named entity recognition constituting the task. In other words, according to the results of the experiment conducted, when there were five dialogue turns, the most ideal entity-inclusion ratio and classification performance are achieved.

### 4.3. Experiment for Snippet Ranking

The XLNet model combines the advantages of the auto-regressive model represented by GPT and the auto-encoder model represented by BERT, achieving state-of-the-art for several natural-language-processing tasks [39]. The previously used BERT has a limitation of up to 512 tokens in sequence length, but XLNet has no limitation, so it can handle large documents. Therefore, it is suitable for implementing the corresponding task, which uses negative sampling and conversational history together as the input. Therefore, we implemented it to rank snippets according to their relevance to the conversation history to select the most appropriate knowledge snippets for response generation [42,43]. To improve the performance of the ranking model, four negative sample application methods were constructed, i.e., all, positive, random, and in-domain, during training, and the results of the ranking experiment performance evaluation are shown in Table 6 below.

Looking at the experimental results, the results using the XLNet model achieved better overall performance than the baseline model [7] implemented through GPT2. Comparing each negative sampling method, the lowest performance among the four methods was obtained as the result of training using all snippets without using domain and entity, which represented the information obtained from the previous task. Here, when it matched domain and entity information, it was composed of only positive samples, and as the result of reducing the number of candidates and using them as knowledge, it improved performance over using the entire sample.

Looking at the experimental results, the results using the XLNet model achieved better overall performance than the baseline model [7] implemented through GPT2. Therefore, we proved that the limitations of the GPT model can be overcome by capturing and learning bidirectional contexts by using an auto-encoder model as well as an auto-regressive model via XLNet. Additionally, comparing the results of each negative sample method ‘all’, the result of snippet ranking without using the domain information obtained from the domain classification and entity information obtained from entity extraction was trained using all snippets, so it had the lowest performance among the four methods. On the other hand, the positive method, which reduced the number of candidates of snippets by constructing knowledge only with positive samples whose domain and entity information were consistent with the previous task, had improved performance compared with the all method using all samples. Therefore, based on the results, the hierarchical structure of the knowledge-selection step proposed in this paper can be justified, and the importance of the task can be confirmed by improving performance using the information obtained as the result of each structure.

Furthermore, the random and in-domain methods, which applied negative sampling in earnest, had better performance than the previous two methods, so negative sampling was effective for ranking. In addition, the need for domain classification and entity extraction tasks could be confirmed through the results of configuring negative samples using the domain and entity information obtained earlier than randomly selecting negative samples among all snippets. Furthermore, it could be inferred that actively using information and knowledge for proper snippet selection helped improve performance through the experimental results below, and that the ultimate goal of this paper, the system, could generate a response based on appropriate knowledge of the user’s question.

In addition, we would like to compare the model conducted in this paper with other state-of-the-art models. Based on a Robustly Optimized BERT Pretraining Approach (ROBERTa-WD) model [44], negative samples were constructed as a process of importance sampling through the ‘k-fold cross-validated style’. Additionally, compared with the method of augmentation through the Topical-Chat and Topical-Chat ASR datasets [13], the model proposed in this paper [45] achieved better performance. Therefore, it could be confirmed through experiments that training without data augmentation by constructing a negative sample using domain and entity information as knowledge, as conducted in this study, was more effective than applying augmentation. In addition, based on the Efficiently Learning an Encoder that Classifies Token Replacements Accurately (ELECTRA) model [46], multi-task learning is applied to extract domain and entity information, and top-three negative sample lists that can cause confusion through training are constructed and compared with the method used [47]. As a result, in this study, a list of negative samples was constructed based on the domain and entity information obtained equally, and better performance was achieved using a larger number of negative samples than the above model.

## 5. Conclusions

To provide appropriate answers or services to users in a task-oriented dialogue system, it is necessary to utilize the knowledge brought by the database or API to generate responses. To overcome the limitation that internally constructed knowledge cannot respond to all user requests, this paper solves it by actively utilizing external unstructured knowledge. Therefore, out of the three stages of unstructured domain knowledge-grounded task-oriented dialogue modeling, the knowledge selection steps were subdivided into domain-classification, entity-extraction, and snippet-ranking tasks to utilize knowledge in the dialogue system. Each task was configured to select an appropriate knowledge snippet by reducing the search range of candidates by using the result obtained from the previous task as knowledge in a hierarchical structure, thereby improving the performance and efficiency of the experiment.

In this paper, several advanced techniques, such as IOB tagging and negative sampling, were successfully implemented for each task by applying them to knowledge selection. In addition, when the negative sampling method, which was constructed using domain and entity information obtained through domain classification and entity extraction task, was finally configured in the snippet-ranking task, it was proven through experiments that it performed well compared with other baseline models and state-of-the-art models.

However, this paper focuses on the knowledge-selection step among the three steps of knowledge-grounded task-oriented conversational modeling, so the first step, the knowledge-seeking turn-detection step, which distinguishes whether knowledge is needed in the conversation turn, and the knowledge-grounded generation step that generates answers based on knowledge about user questions is not implemented. Therefore, with the knowledge gained regarding the hierarchical structure of knowledge selection claimed in this paper, we will conduct future work on an end-to-end process that generates an eloquent answer to a user’s question.

## Figures and Tables

**Figure 1 sensors-23-00685-f001:**
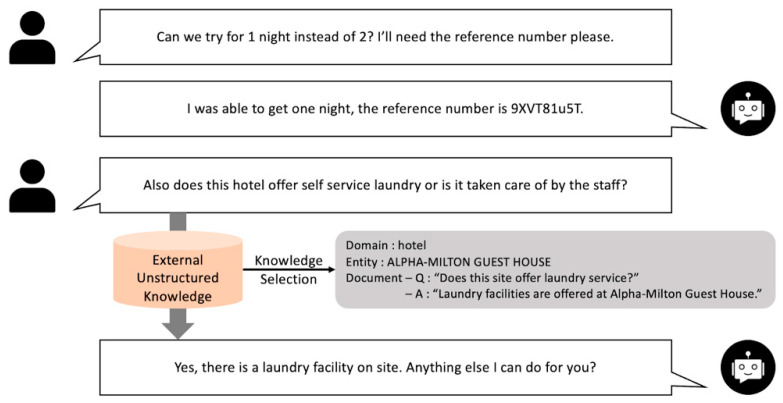
Example of the proposed knowledge-grounded task-oriented dialogue system.

**Figure 2 sensors-23-00685-f002:**
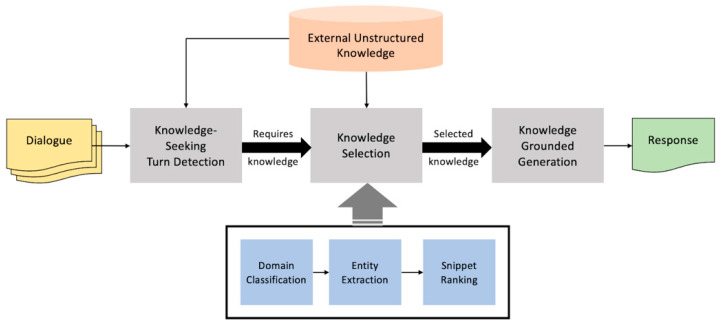
Baseline architecture for the knowledge-grounded task-oriented dialogue system.

**Figure 3 sensors-23-00685-f003:**
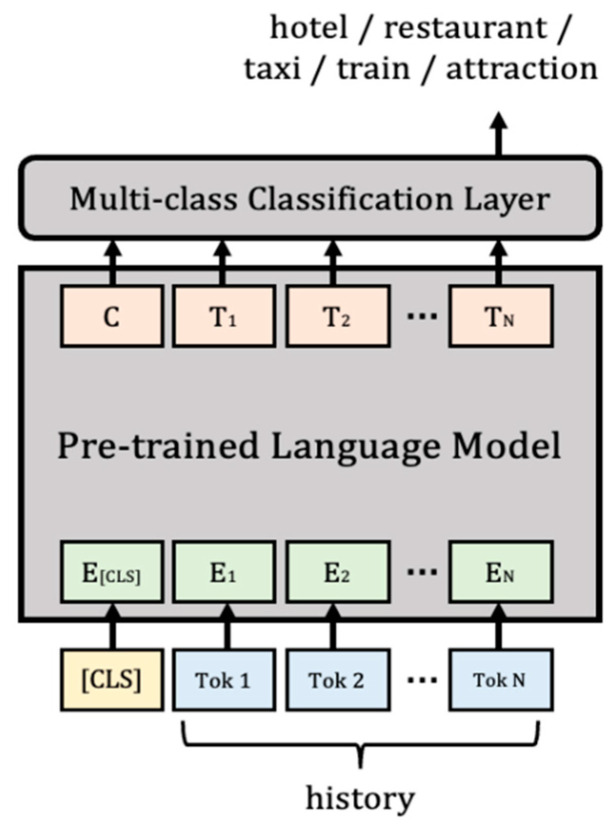
Domain-classification model.

**Figure 4 sensors-23-00685-f004:**
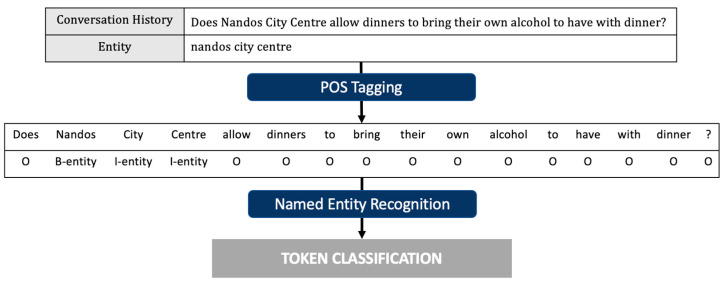
Overall process of the entity-extraction task with POS tagging.

**Figure 5 sensors-23-00685-f005:**
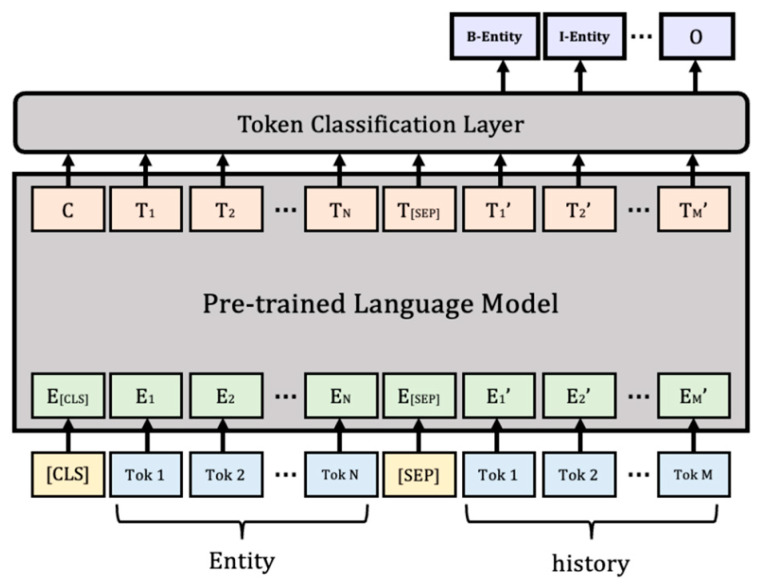
Entity-extraction model.

**Figure 6 sensors-23-00685-f006:**
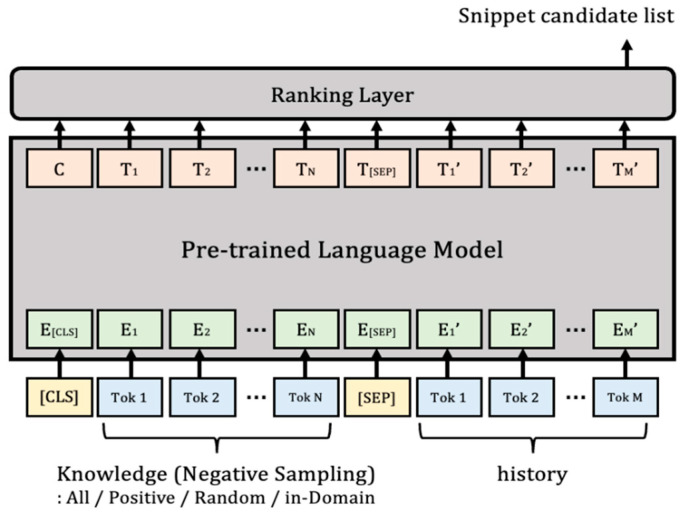
Snippet-ranking model.

**Table 1 sensors-23-00685-t001:** Description of MultiWOZ dataset.

Dataset Type	# Dialogs	# Total Turns	# Knowledge Seeking Turns	# Knowledge Snippets	Domain Type
Train	7190	71,348	1,9184	2900	Hotel, restaurant, taxi, train
Valid	1000	9663	2673	2900	Hotel, restaurant, taxi, train
Test	977	4181	1981	12,039	Hotel, restaurant, taxi, train, attraction

**Table 2 sensors-23-00685-t002:** Domain-classification result with different models and the number of dialogue turns.

Model	# Dialogue Turn	Precision	Recall	F1-Score
BERT	1	0.6611	0.7436	0.6922
	3	0.7078	0.8092	0.7523
	5	0.7304	0.8269	0.7706
	10	0.7378	0.8339	0.7777
	15	0.7380	0.843	0.7846
DistilBERT	1	0.6609	0.7415	0.6906
	3	0.7115	0.8077	0.7523
	5	0.7339	0.8349	0.7778
	10	0.7316	0.8390	0.7802
	15	0.7406	0.8455	0.7870

**Table 3 sensors-23-00685-t003:** Classification accuracy for each domain according to the number of dialogue turns.

# Dialog Turn	Classification Accuracy				
	Hotel	Train	Restaurant	Taxi	Attraction
1	514	337	468	150	0
3	548	344	534	174	0
5	540	343	601	170	0
10	562	344	573	183	0
15	572	345	576	182	0
	574	347	611	185	264

**Table 4 sensors-23-00685-t004:** Percentage of entities contained for each dataset according to the number of dialog turns.

# Dialog Turn	Ratios of Entities		
	Training Set	Validation Set	Test Set
1	0.1810	0.1900	0.2188
3	0.3494	0.3199	0.4669
5	0.445	0.4141	0.5931
10	0.5166	0.4718	0.6628
15	0.9561	0.4856	0.9234

**Table 5 sensors-23-00685-t005:** Entity-extraction results with different models and numbers of dialogue turns.

Model	# Dialog Turns	Precision	Recall	F1-Score
BERT	1	0.920	0.921	0.919
	3	0.760	0.758	0.755
	5	0.687	0.678	0.678
	10	0.623	0.611	0.609
	15	0.59	0.584	0.58
DistilBERT	1	0.929	0.933	0.930
	3	0.836	0.833	0.832
	5	0.755	0.755	0.753
	10	0.705	0.708	0.703
	15	0.682	0.683	0.678

**Table 6 sensors-23-00685-t006:** Snippet ranking results with different negative samples.

Model	MRR@5	Recall@1	Recall@5	Average
Baseline (gpt2)	0.7263	0.6201	0.8772	0.7412
ROBERTA-WD (importance sampling + data augmentation)	0.9349	0.8941	0.9835	0.9375
ELECTRA (negative sampling + multi-task learning)	0.9372	0.9117	0.9665	0.9385
XLNet + all	0.8716	0.7905	0.9824	0.8815
XLNet + positive	0.9240	0.8819	0.9899	0.9319
XLNet + random	0.9324	0.8869	0.9955	0.9383
XLNet + in-domain	0.9469	0.9056	0.9934	0.9486

## Data Availability

Not applicable.

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
