# Peer review of "A Knowledge-Grounded Task-Oriented Dialogue System with Hierarchical Structure for Enhancing Knowledge Selection"

_sensors, 2023, doi:10.3390/s23020685_

Round 1
Reviewer 1 Report
The paper describes development of dialogue systems with hierarchical structure. As the main contribution the authors highlight decomposition of the knowledge selection step to Domain classification, Entity extraction, and Snippet ranking.
Comments:
Abstract: “three steps” are mentioned but not explained
Line 45 and further: Please explain “conversational turn” term
Abbreviations must be explained before usage (e.g. IOB is not decrypted)
Related work could be extended to better highlight the difference and impact of the proposed approach
Lines 211-213: “amount of conversational history to be used for input is organized differently into {1, 3, 5, 10, 15}” needs to be clarified
Lines 220: it is not obvious that NER can fully solve the problem of entity extraction, the dialogue does not always refer to named entities, please clarify;
I-entity, B-entity need to be introduced and briefly explained
Sections 4.1, 4.2, 4.3 seem not to be detailed or clear enough to fully explain the contributions in the methods proposed by the authors; maybe adding pseudocode could help
Experimental part (Section 5) should better reflect own findings and contribution; now it is not fully clear how this is highlighted with results on known datasets and NNs. Some explanation for used and preferred models is needed (DistilBERT, XInet), now they are only briefly mentioned.
Author Response
Author’s Reply to the Reviewer Report
- Journal: Sensors
- Manuscript ID: sensors-2077943
- Title: A Knowledge-grounded Task-oriented Dialogue system with hierarchical structure for enhancing knowledge selection
- Section: Sensors and Robotics
- Special Issue: Artificial Intelligence for Decision Making
Dear Editor:
Thank you very much for your time and efforts spent in reviewing this paper. We could develop our paper in more detail thanks to the in-depth review on it by reviewers. It is our pleasure that reviewers are interested in the theme of the paper and the proposed methods. Also, we could improve our paper based on the reviewer’s useful suggestions and comments. Our answers to those comments by the reviewer are as below
Reviewer #1
Comment 1.1: Abstract: “three steps” are mentioned but not explained
Answer 1.1: The explanation of the three steps was added to the abstract line 11-13. “So, this paper constructs a 3-step Knowledge-grounded Task-oriented Dialogue system with knowledge-seeking turn detection, knowledge selection, and knowledge-grounded generation.”
Comment 1.2: Line 45 and further: Please explain “conversational turn” term
Answer 1.2: ‘Conversational turn’ means each turn in the multi-turn conversation between the user and the system, so an additional description was supplemented to the corresponding part.
Comment 1.3: Abbreviations must be explained before usage (e.g. IOB is not decrypted)
Answer 1.3: The original full term of abbreviations was described as an IOB (inside-outside-beginning) tagging, and all other abbreviations used in the paper were also included.
Comment 1.4: Related work could be extended to better highlight the difference and impact of the proposed approach
Answer 1.4: In selecting external unstructured knowledge among the detailed tasks of the knowledge selection proposed in this paper, we added a description of the Named Entity Recognition method, a representative information retrieval method used together to successfully perform entity extraction, the most important task. Therefore, we tried to increase the persuasive power by explaining the background knowledge of what this paper claims and emphasizing the differences from other studies.
Comment 1.5: Lines 211-213: “amount of conversational history to be used for input is organized differently into {1, 3, 5, 10, 15}” needs to be clarified
Answer 1.5: In the conversation composed of multi-turns, it was judged that the explanation of constructing the input by varying the number of conversational turns was insufficient, so it was revised as follows. “In addition, to find out where important information is mainly located to understand the context of the conversation, the experiment is conducted by organizing the amount of conversation history to be used as input in multi-turn conversations between user and the system of up to 15 turns according to {1, 3, 5, 10, 15}.”
Comment 1.6: Lines 220: it is not obvious that NER can fully solve the problem of entity extraction, the dialogue does not always refer to named entities, please clarify;
Answer 1.6: NER is a technique for extracting the desired named entity from text, which is suitable for the entity extraction task to obtain the entity information contained in the conversation history. A detailed description related to NER was described in the corresponding part, and additionally, [2. Related Work] described the necessary description along with the background knowledge to successfully solve the entity extraction.
Comment 1.7: I-entity, B-entity need to be introduced and briefly explained
Answer 1.7: In the same way as answer 1.6, a detailed description of the tag was written, and Figure 4 was added to aid understanding.
Comment 1.8: Sections 4.1, 4.2, 4.3 seem not to be detailed or clear enough to fully explain the contributions in the methods proposed by the authors; maybe adding pseudocode could help
Answer 1.8: Sections 4.1, 4.2, 4.3 were revised to 3.1, 3.2, 3.3, and the description was modified to emphasize the contributions of the method claimed in this paper. However, the three tasks solve the problem by adding the desired layer for each task to the pre-trained language model. Therefore, since the deep learning-based model, which is complex black box model, is used to obtain the desired output, I think a figure or written description indicating the process in detail is more appropriate than writing a pseudocode. Therefore, a supplementary explanation was added to the corresponding task along with Figure 4.
Comment 1.9: Experimental part (Section 5) should better reflect own findings and contribution; now it is not fully clear how this is highlighted with results on known datasets and NNs. Some explanation for used and preferred models is needed (DistilBERT, XInet), now they are only briefly mentioned.
Answer 1.9: A detailed description of each of the pre-trained language models BERT, DistilBERT, and XLNet used for the three large tasks of [4. Experiments], and an explanation based on the reasons and advantages of using the model for each task were added. In addition, in the comparative experiment section in Table 6, an additional explanation of the excellence of the method proposed in this paper was written based on objective facts to compensate for the deficiencies.

Reviewer 2 Report
The paper contribute to the body of knowledge and deals with very actual thema to intend to improve the performance of the Knowledge-grounded Task-oriented dialogue system by applying detailed tasks to the knowledge selection stage to efficiently utilize related domain knowledge such as the type of FAQ and customer review called external unstructured knowledge. But the paper is not technically sound:
1.Abstract
1.1. Abstract must content concise review of all the content of the the article and short but clear explaintation of it's concrete contribution to scientific knowladge.
2. Introduction and Related work sections are placed in the right place but:
2.1 INTRODUCTION
Presented problem which is considered in this paper gaps dealing with specific hypotheses being tested so the authors did not follow the clearly given instructions of journal:
„The introduction should briefly place the study in a broad context and highlight why it is important. It should define the purpose of the work and its significance, including specific hypotheses being tested. The current state of the research field should be reviewed carefully and key publications cited. Please highlight controversial and diverging hypotheses when necessary. Finally, briefly mention the main aim of the work and highlight the main conclusions. Keep the introduction comprehensible to scientists working outside the topic of the paper.“
2.2. RELATED WORK
Must includes a larger number of existing 13 given references from the world literature which must be described in more detail and which normaly deals with subject of this paper, in order to show the topicality of the subject matter of this paper.
3. Sections 3.Dataset and 4.Proposed method
These sections deal practically with the Matherial and Method and maybe it will be better name for common chapter in which existing two could be suitable subchapters 3.1 Dataset and 3.2 Proposed method and thus the organization of the paper would make the content of the article more accessible to the reader.
4. Section 5.Experiments
This section of manuscript is lacking with comparative evaluation of the proposed algorithm with other already existing as well as it's limitations and could have name Results and finding for example or ....
5. The manuscript is lacking with presented future work in section Conclusion.
6. It is inadmissible for an article intended to be published in an eminent journal such as Sensors to refer to only 37 references, especially only 13 of them in section Related works.
Author Response
Author’s Reply to the Reviewer Report
- Journal: Sensors
- Manuscript ID: sensors-2077943
- Title: A Knowledge-grounded Task-oriented Dialogue system with hierarchical structure for enhancing knowledge selection
- Section: Sensors and Robotics
- Special Issue: Artificial Intelligence for Decision Making
Dear Editor:
Thank you very much for your time and efforts spent in reviewing this paper. We could develop our paper in more detail thanks to the in-depth review on it by reviewers. It is our pleasure that reviewers are interested in the theme of the paper and the proposed methods. Also, we could improve our paper based on the reviewer’s useful suggestions and comments. Our answers to those comments by the reviewer are as below
Reviewer #2
Comment 2.1.1: [1. Abstract]
Abstract must content concise review of all the content of the article and short but clear explanation of it’s concrete contribution to scientific knowledge.
Answer 2.1.1: An overview of the knowledge-grounded Task-oriented dialogue system and hierarchical structure that this paper intends to propose is added to [Abstract]. As a result, the overall explanation of the paper was rewritten accurately but briefly by adding an explanation of the contribution made by this paper.
Comment 2.2.1: [2.1 Introduction] Presented problem which is considered in this paper gaps dealing with specific hypotheses being tested so the authors did not follow the clearly given instructions of journal: “The introduction should briefly place the study in a broad context and highlight why it is important. It should define the purpose of the work and its significance, including specific hypotheses being tested. The current state of the research field should be reviewed carefully and key publications cited. Please highlight controversial and diverging hypotheses when necessary. Finally, briefly mention the main aim of the work and highlight the main conclusions. Keep the introduction comprehensible to scientists working outside the topic of the paper.”
Answer 2.2.1: As the comment prepared by [1. Introduction], it was revised to emphasize why the hierarchical structure of knowledge selection is important along with the background knowledge of the method proposed in this paper. In addition, the necessity of the study was described based on the explanation of other major studies, and it was revised and prepared to reveal the main goals well.
Comment 2.2.2: [2.2 Related Work] Must includes a larger number of existing 13 given references from the world literature which must be described in more detail and which normally deals with subject of this paper, in order to show the topicality of the subject matter of this paper.
Answer 2.2.2: In selecting external unstructured knowledge among the detailed tasks of the knowledge selection proposed in this paper, we added a description of the Named Entity Recognition method, a representative information retrieval method used together to successfully perform entity extraction, the most important task. Therefore, we tried to increase the persuasive power by explaining the background knowledge of what this paper claims and emphasizing the differences from other studies.
Comment 2.3: [3. Sections: 3. Dataset and 4. Proposed Method] These sections deal practically with the Material and Method and maybe it will be better name for common chapter in which existing two could be suitable subchapters 3.1 Dataset and 3.2 Proposed method and thus the organization of the paper would make the content of the article more accessible to the reader.
Answer 2.3: Section 3 was renamed to a proposed method, changed 3.1-3.3 as a description of three hierarchical tasks of the existing proposed method and modified to 3.4 Dataset.
Comment 2.4: [4. Section: 5. Experiments] This section of manuscript is lacking with comparative evaluation of the proposed algorithm with other already existing as well as it's limitations and could have name Results and finding for example or ....
Answer 2.4: For the three tasks constituting [4. Experiments], a detailed description of the pre-trained language model implementing each task was described, and an explanation of advanced techniques such as negative sampling, which was additionally used together to improve performance, was also written. Furthermore, the need for hierarchical structure is also confirmed based on comparative experiments with other studies or using different models and methods to justify the techniques claimed in this paper.
Comment 2.5: The manuscript is lacking with presented future work in section Conclusion.
Answer 2.5: At the end of [5. Conclusion], we added an explanation that we will cover knowledge-seeking turn detection and knowledge-grounded generation tasks among the three steps of knowledge-grounded task-oriented conversational modeling that this paper has not covered.
Comment 2.6: It is inadmissible for an article intended to be published in an eminent journal such as Sensors to refer to only 37 references, especially only 13 of them in section Related works.
Answer 2.6: A reference was added to the Related Work to supplement the description of a total of 20 other studies and the background knowledge of the various techniques and structures used in this paper. Furthermore, the overall number of papers was increased to 48 by adding a reference.

Round 2
Reviewer 1 Report
I thank the authors for addressing my earlier comments. I believe the manuscript has been improved enough to be accepted for publication.
Author Response
Dear Reviewer:
Thank you very much for your time and efforts spent in reviewing this paper. We could develop our paper in more detail thanks to the in-depth review on it by reviewers. It is our pleasure that reviewers are interested in the theme of the paper and the proposed methods. Also, by making detailed corrections such as adding English spelling and abbreviation description, we could improve our paper based on the reviewer’s useful suggestions and comments.
Reviewer 2 Report
I can remark that the authors accepted all my suggestions and corrected the paper according to them. So, now I can happily support publication of revised manuscript in the journal Sensors.
Having in mind jornal's guide for authors it is necessery to remove the abbreviation API from the Abstract of paper and also define before using the abbrevations in the manuscript as it are AI, GPT2,TF-IDF ...
Author Response
Dear Reviewer:
Thank you very much for your time and efforts spent in reviewing this paper. We could develop our paper in more detail thanks to the in-depth review on it by reviewers. It is our pleasure that reviewers are interested in the theme of the paper and the proposed methods. Also, by removing the abbreviation API from the Abstract of paper and adding explanations for abbreviations such as GPT2 and TF-IDF, we could improve our paper based on the reviewer’s useful suggestions and comments.